# Synergy between Indoloquinolines and Ciprofloxacin: An Antibiofilm Strategy against *Pseudomonas aeruginosa*

**DOI:** 10.3390/antibiotics10101205

**Published:** 2021-10-04

**Authors:** Emilie Charpentier, Ludovic Doudet, Ingrid Allart-Simon, Marius Colin, Sophie C. Gangloff, Stéphane Gérard, Fany Reffuveille

**Affiliations:** 1EA 4691 Biomatériaux et Inflammation en Site Osseux (BIOS), UFR Pharmacie, Université de Reims Champagne-Ardenne, SFR Cap Santé (FED 4231), 51097 Reims, France; emilie.charpentier@univ-reims.fr (E.C.); marius.colin@univ-reims.fr (M.C.); sophie.gangloff@univ-reims.fr (S.C.G.); 2Institut de Chimie Moléculaire de Reims (ICMR-UMR CNRS 7312), UFR Pharmacie, Université de Reims Champagne-Ardenne, 51097 Reims, France; ludovic.doudet@univ-reims.fr (L.D.); ingrid.allart-simon@univ-reims.fr (I.A.-S.); stephane.gerard@univ-reims.fr (S.G.)

**Keywords:** biofilm, synergy, indoloquinoline, *Pseudomonas aeruginosa*

## Abstract

Antibiotic treatments can participate in the formation of bacterial biofilm in case of under dosage. The interest of indoloquinoline scaffold for drug discovery incited us to study the preparation of new indolo [2,3-*b*]quinoline derivatives by a domino radical process. We tested the effect of two different “indoloquinoline” molecules (Indol-1 and Indol-2) without antimicrobial activity, in addition to ciprofloxacin, on biofilm formation thanks to crystal violet staining and enumeration of adhered bacteria. This association of ciprofloxacin and Indol-1 or Indol-2 attenuated the formation of biofilm up to almost 80% compared to ciprofloxacin alone, or even prevented the presence of adhered bacteria. In conclusion, these data prove that the association of non-antimicrobial molecules with an antibiotic can be a solution to fight against biofilm and antibiotic resistance emergence.

## 1. Introduction

The emergence of antibiotic resistance and antimicrobial tolerant biofilm represent a major threat of incurable infectious diseases. Fighting them must constitute a priority where the best strategy is to study how to prevent their induction and formation.

Biofilms are communities of bacteria irreversibly attached to a surface or to each other and are embedded in a matrix of various components, with completely different metabolisms from free-living bacteria [1,2]. Scientists estimate that around 80 % of bacterial infections involve biofilms [3]. Unfortunately, these super structures are tolerant to antibiotics and immune system, representing a high concern for public health. Bacteria respond very quickly to their environment in order to survive, and biofilms constitute an excellent stress response and survival strategy [4]. For example, several studies have shown that antibiotic treatments can induce biofilm formation [5,6]. The situation can then worsen since the proximity of bacteria in the biofilm also induces the emergence of antibiotic resistance due to horizontal gene transfer [7].

In the clinical field, the fluoroquinolone ciprofloxacin possesses an excellent broad spectrum antimicrobial activity and pharmacokinetic properties, presents few side effects and is used as therapy against urinary tract infections, respiratory infections and bone and soft tissue infections [8], caused by both Gram-positive or Gram-negative bacteria. Ciprofloxacin targets type II DNA topoisomerases (DNA gyrases) by inhibiting synthesis of bacterial mRNAs, stopping DNA replication. The Minimal Inhibitory Concentration (MIC) is low, in general less than 1 µg/mL: 0.25–0.5 µg/mL for *Staphylococcus aureus* and *Pseudomonas aeruginosa* (*P. aeruginosa*) but 0.08 µg/mL for *Escherichia coli* [9], for example. The World Health Organization classified ciprofloxacin on the essential drug list and it remains the most important treatment against *P. aeruginosa* infections [10]. Unfortunately, sub-inhibitory concentration of ciprofloxacin induced biofilm formation, which is itself increasing antibioresistance emergence. Indeed, all bacteria developed resistance to ciprofloxacin due to inappropriate use [10].

In parallel, alkaloid possessing indoloquinoline scaffolds are important heterocyclic compounds in medicinal chemistry [11]. *N*-methylbenzofuro[3,2-*b*]quinolones, for example, were recently synthesized and tested for their activity against methicillin-resistant *S. aureus* [12] while indolo[2,3-*b*]quinoline analogs also exhibited anti-MRSA activity [13]. Ali et al. [14] also revealed the indolo[2,3-*b*]quinolines strong activity against *Aspergillus niger* and *Bacillus subtilis*. To our knowledge, antibiofilm activity of these compounds or the emergence of resistance against them have not been tested.

To prevent both the emergence of resistance and the formation of biofilm, a powerful strategy would be to use combinations of antibacterial molecules and antibiofilm ones. This approach has already been tested with promising results, underlining the importance to independently present the two types of activities on different molecules [15,16,17,18]. In this study, we investigated on two indoloquinoline derivatives as antibiofilm candidates, in combinations with a well-known antibiotic, the ciprofloxacin, at MIC and sub-MIC to prevent *P. aeruginosa* biofilm formation.

## 2. Results

### 2.1. Preparation of Indoloquinoline Derivatives

As part of our ongoing program dealing with the synthesis of complex indole-heterocycles by tandem reactions, we previously investigated whether a 5-exo-trig cyclisation could be innovatively combined to a radical promoted Smiles rearrangement [19]. A recent study established and validated a computational tool to develop mechanistic investigations enabling the synthesis of new scaffolds of therapeutic interest by this approach [20]. Because of the interest of indoloquinoline scaffold for drug discovery [21], we studied the preparation of indolo[2,3-b]quinoline derivatives by a domino process including radical Smiles rearrangement [22,23,24]. The key iodo-precursor could be synthesized by a convergent approach and with this precursor in hand we next turn to our domino process using 2,2-azocyclohexanecarbonitrile (ACCN), a liposoluble radical initiator, and tris(trimethylsilyl)silane (TTMSS) as reducing agent. These radical conditions allowed formation of a rearranged product, which could efficiently be converted to the targeted compounds **1** and **2** possessing tetracyclic scaffold (Figure 1). In the context of another study, none of the compounds (10 µg/mL Indol-1 and Indol-2) exhibited statistically relevant cytotoxic activity on circulating peripheral blood mononuclear cells.

### 2.2. Impact of Ciprofloxacin on Biofilm Formation

Testing antimicrobial activity of these compounds, we did not detect any effect on planktonic growth or biofilm formation when Indol-1 and Indol-2 were used without ciprofloxacin (Appendix A).

In minimal medium, the MIC of ciprofloxacin against *P. aeruginosa* strain was 0.63 μg/mL but we noticed a planktonic growth decline starting with 0.04 µg/mL (Figure 2). The sub-MIC concentrations increased biofilm formation compared to non-treated condition by 2.1 to 2.6 fold-changes (Mann-Whitney test; *p* < 0.05). Moreover, biofilm biomass at MIC was only reduced by 43% compared to non-treated biofilm condition despite an absence of detectable planktonic growth.

### 2.3. Combination of Ciprofloxacin and Indoloquinolines to Decrease Biofilm Biomass

After observing that ciprofloxacin MIC did not prevent biofilm formation, the addition of an indoloquinoline as supplementary molecule was tested. Indol-1 and Indol-2 were tested at 1.25 and 2.5 µg/mL. Regardless of the indoloquinoline nature or concentration, its addition did not affect the ciprofloxacin MIC (Figure 3a,b). A non-significant decrease of biofilm biomass of 57% was observed when *P. aeruginosa* culture was treated with ciprofloxacin MIC and 2.5 µg/mL of Indol-1 compared to ciprofloxacin MIC treatment alone (Figure 3a), i.e., a significant drop of 88% compared to the untreated biofilm (Mann-Whitney test; *p* < 0.05). Interestingly, a decrease of 79 % and 72 % of biofilm quantity was noticed with the addition of 1.25 and 2.5 µg/mL of Indol-2, respectively, compared to ciprofloxacin MIC alone, i.e., almost 90% of biofilm loss compared to non-treated control (Mann–Whitney test; *p* < 0.05) (Figure 3b).

Indol-1 at 1.25 or 2.5 µg/mL combined with MIC of ciprofloxacin led to a 53% and 37% decrease of the adherent bacteria quantity, respectively (Figure 3c). A drop of 62% of attached bacteria was also observed with 1.25 µg/mL of Indol-2 combined with MIC of ciprofloxacin. Surprisingly, no adherent bacteria were detected after 2.5 µg/mL Indol-2 + MIC of ciprofloxacin treatment after 24 h (Mann-Whitney test; *p* < 0.05).

### 2.4. Combination of Ciprofloxacin and Indoloquinoline to Decrease Adherent Bacteria Quantity

As sub-MIC of ciprofloxacin strongly induced biofilm formation, we chose to combine ciprofloxacin at MIC divided by 2 or 4, with Indol-1 or Indol-2 (Figure 4).

Planktonic growth at MIC/2 and MIC/4 was estimated around 0.1 and 0.4 of absorbance, respectively, and did not change in the presence of indoloquinoline, except a non-significant decrease with the addition of 2.5 µg/mL of Indol-2. MIC/4 combined with 2.5 µg/mL of Indol-1 was the only combination to significantly reduce biofilm biomass, suggesting a higher impact of lower concentrations of Indol-1 against *P. aeruginosa* biofilm formation.

To go further, we evaluated the quantity of live adherent bacteria in the presence of these molecules. At MIC/2 of ciprofloxacin, the addition of 2.5 µg/mL of Indol-1, 1.25 µg/mL of Indol-2 or 2.5 µg/mL of Indol-2 led to a significant decrease of 78%, 93% and 100 % of adherent cells, respectively (Mann–Whitney test; *p* < 0.05). Non-significant results were observed at MIC/4, regardless of Indol concentration, although adherent bacteria quantities still tended to decrease in the presence of Indol-1 or 2.

## 3. Discussion

Ciprofloxacin is a fluoroquinolone, which is currently used in the treatment of *P. aeruginosa* infections, even if some resistances could be detected and most probably linked to the establishment of *P. aeruginosa* biofilms. Therefore, it now appears crucial to investigate on alternative solutions able to avoid biofilm formation, and thus to simplify infections treatments.

Here, we started by determining the minimum inhibitory concentration of ciprofloxacin in a specific culture medium. The latter is a so-called “minimum” medium because the nutrient deficiency participates in the formation of biofilm [25,26] and is more representative of bacterial starvation condition during infection than classical culture media [27]. Under these culture settings, we observed the danger of using MIC and/or sub-MIC concentrations of ciprofloxacin. Indeed, biofilm formation of *P. aeruginosa* was not completely inhibited, even though a treatment at MIC of ciprofloxacin. Indeed, while no planktonic growth was detected, 60% of the biofilm biomass remains compared to the untreated *P. aeruginosa* biofilm culture (control), underlining the bacterial tolerance under biofilm mode. This observation correlates with previous studies highlighting that the minimal biofilm inhibitory concentration was often evaluated at higher value than MIC [28,29,30]. Thus, antibiotic concentrations used in clinic should be higher than MIC to avoid biofilm development but this approach is rapidly limited by antibiotic cytotoxicity. The presence of a biofilm at MIC is worrying because of its high tolerance to antimicrobial and immune system. In addition, bacteria released from biofilm colonizes other sites leading to secondary infections [31,32]. In this study, we confirmed that sub-MIC concentration lead to increase *P. aeruginosa* biofilm formation as previously described in literature [33,34,35]

To avoid this problem, combining multiple molecules with different targets is one hopeful strategy. In the present study, the use of new indoloquinoline molecules in addition of ciprofloxacin against *P. aeruginosa* biofilm was tested. Interestingly, addition of Indol-1 or Indol-2 in low concentrations (2.5 µg/mL) led to a decrease of adhered bacteria quantities. The most promising result was the combination of ciprofloxacin at MIC and Indol-2 at 2.5 µg/mL, which allowed the complete inhibition of bacterial adhesion. However, biofilm was still detected by crystal violet staining at that condition. In fact, crystal violet staining reflects all biofilm biomass (live and dead bacteria, biofilm matrix) whereas adhered cell counting method allowed only the estimation of live bacteria in biofilm. Thus, the measured amount of crystal violet staining could reflect the presence of biofilm matrix and dead bacteria alone [36].

We noticed that the combinations of indoloquinolines with only MIC/2 of ciprofloxacin were effective to avoid bacterial adhesion. Especially for compound Indol-2, we observed only 7% of adhered bacteria under a treatment of 1.25 µg/mL Indol-2 and CMI/2 ciprofloxacin compared to ciprofloxacin alone. Furthermore, none adhered bacteria were detected under a 2.5 µg/mL of Indol-2 combined with MIC/2 of ciprofloxacin. These results appear very promising as they suggest the potential abilities of indoloquinolines to counteract pro-biofilm effect of ciprofloxacin at sub-MIC. Surprisingly, only the combination of 2.5 µg/mL of Indol-1 to MIC/4 of ciprofloxacin has shown a significant decrease in amount of biofilm stained by crystal violet (drop of 75% compared to MIC/4 treated *P. aeruginosa*). However, we only observed a tendency of decrease in number of live adherent bacteria under this combination. This result could reveal that the decrease in crystal violet stain amount was caused by a decrease in biofilm matrix or dead bacteria. Such an action may suggest that Indol-1 low concentration treatment would disturb biofilm matrix and decrease its measurable quantity. Thus, investigations will be needed to appreciate the impact of this specific combination treatment on matrix biofilm, as well as on metabolic activity of bacteria within the biofilm. Due to this purpose, the XTT-menadione assay could be used to decipher the mechanism of action involved. Moreover, indoloquinoline-ciprofloxacin combined drugs will have to be tested for their biofilm eradication capacity. Another approach will be to combine Indol-1 and Indol-2 associated together with ciprofloxacin, in order to improve anti-biofilm effect.

## 4. Materials and Methods

### 4.1. Preparation of Indoloquinoline Derivatives

After preparation of key iodo-precursor by coupling reaction of a sulfonamide moiety on an hemiacid-amide partner, the general procedure for the domino radical reaction used a solvent degassed using standard Schlenk techniques, followed by bubbling of argon for 30 min. The substrate and the reducing agent were added by syringe, and immediately afterward the solid initiator was introduced. The solution was then heated up to reflux. After completion of the reaction (TLC monitoring) and cooling to room temperature, the crude mixture was extracted with acetonitrile. The crude product was purified by flash column chromatography on silica gel to afford pure compound Indol-1. Compound Indol-2 is obtained by subsequent *N*-methylation [19]. Molecules were dissolved in DMSO at 1 mg/mL and then diluted in bacterial culture media (to 1/400 for 2.5 µg/mL concentration and 1/800 for 1.25 µg/mL concentration). In each experiment, a solution with the same amount of DMSO but without any molecules was prepared and tested as a control.

### 4.2. Bacterial Strains and Culture Media

*P. aeruginosa* CIP 82.118 or ATCC 9027 is a referent strain cited in European Pharmacopeia. Bacterial strains were cultivated overnight in nutrient medium. A minimal medium (MM) (62 mM potassium phosphate buffer, pH 7.0, 7 mM (NH_4_)_2_SO_4_, 2 mM MgSO_4_, 10 μM FeSO_4_) containing 0.4% (*w*/*v*) glucose and 0.1% (*w*/*v*) casamino acids, was then used to favor biofilm formation [27].

### 4.3. Biofilm Formation

A bacterial culture of 18 h (at stationary phase) was diluted at 1/100 in MM. Then, 500 µL of this dilution were distributed in each well of a 48-well microtiter plate for Crystal violet staining and of a 24-well microtiter plate for counting adhered bacteria. After 24 h incubation, the planktonic growth was evaluated by measuring the absorbance at 600 nm (results are expressed with the subtraction of the blank: medium without bacteria). Biofilm formation was evaluated by Crystal violet staining and counting of adhered bacteria.

### 4.4. Crystal Violet Staining (Biofilm Evaluation)

As previously described, biofilm biomass was evaluated by crystal violet staining [27]. After discarding medium containing planktonic bacteria and three gentle washes, 500 µL of 0.2% of crystal violet was used to stain biofilm for 20 min. After another washing, 500 µL of 95% ethanol was added to each well. The amount of biofilm was quantified by measuring the absorbance at 595 nm (results are expressed with the subtraction of the blank: medium without bacteria). Experiments were done at least with three different overnight cultures at three independent times (9 repeats).

### 4.5. Counting of Adhered Bacteria

Live adhered bacteria number was evaluated thanks to the use of ultrasounds to detach bacteria. As described above, a 1/100 diluted overnight culture in MM was distributed in 24-well plates in presence of a plastic lamella (ThermanoxTM, Nunc, Denmark) at the bottom of the well. After 24 h of incubation, the lamella was washed to eliminate planktonic bacteria, and transferred to a 15 mL Falcon tube containing 2 mL of minimal media. Bacteria were then detached by exposing the sample to 5 min of ultrasound (40 kHz). A volume of 100 µL from serial dilutions was plated on nutrient agar plates before and after ultrasounds to determine the quantity of attached bacteria.

### 4.6. Statistical Methods

The statistical significance of the results was assessed using non-parametric analysis with pairwise tests. The exact non-parametric Wilcoxon Mann–Whitney test for independent samples was used (StatXact 7.0, Cytel Inc., Cointrin, Switzerland). Differences were considered significant at *p* < 0.05.

## 5. Conclusions

In conclusion, non-antimicrobial molecules indoloquinolines (Indol-1 and Indol-2) could be used in combination with antimicrobial ciprofloxacin to avoid its worrying side effect, which is the induction of *P. aeruginosa* biofilm formation. This proof of concept is promising for many reasons. First, the combination of non-antimicrobial molecule and antibiotic might reduce the emergence of tolerance mechanism like biofilm. Second, the combination did not show any antagonistic impact on the ciprofloxacin antimicrobial activity. Third, even if ciprofloxacin treatment leads to sub-MIC concentrations in situ, the Indol-1 and Indol-2 indoloquinolines still inhibit live bacteria attachment in biofilm. Altogether, it will prevent biofilm formation during antibiotic treatment, and, in consequence, may help to decrease the number of therapeutic fails.

## Figures and Tables

**Figure 1 antibiotics-10-01205-f001:**
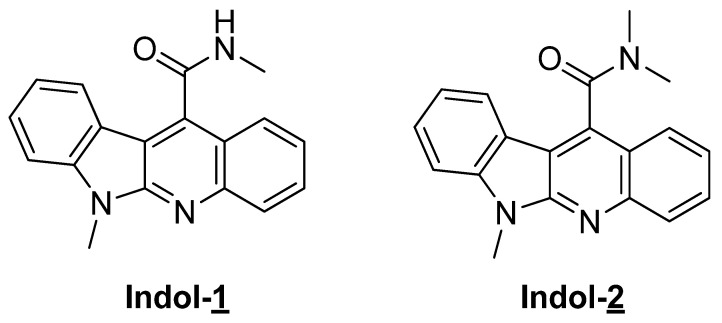
Indolo [2,3-b] quinolines.

**Figure 2 antibiotics-10-01205-f002:**
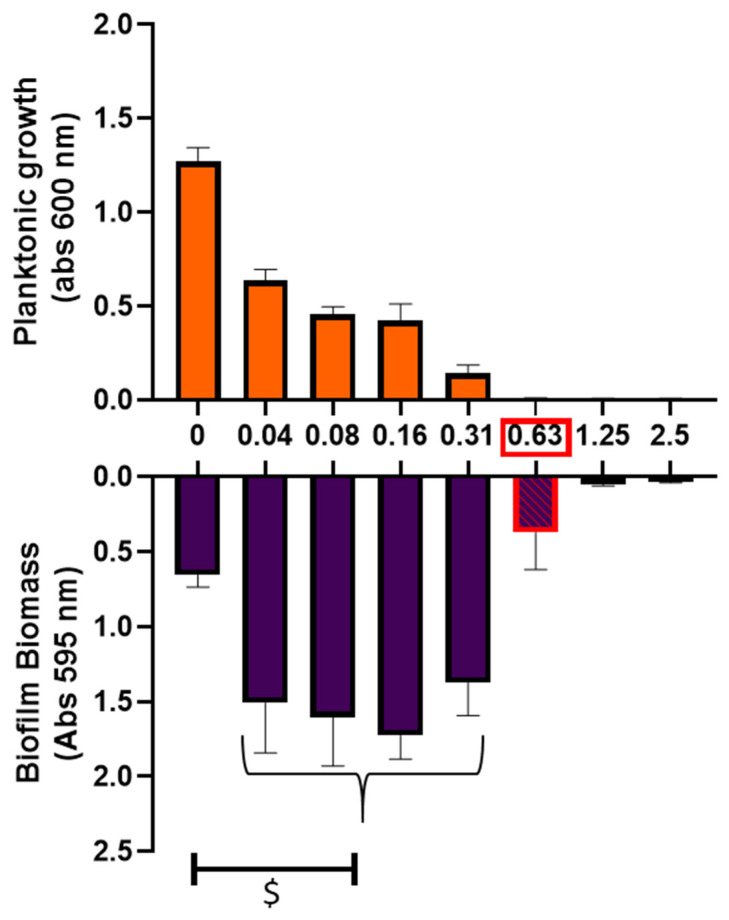
Ciprofloxacin treatment of *P. aeruginosa*. The upper graph represents planktonic growth after 24 h incubation at a range of ciprofloxacin concentrations (absorbance at 600 nm) and the lower graph represents biofilm biomass under the same range of ciprofloxacin concentration treatment after 24 h. Ciprofloxacin MIC appears in red square. Mann–Whitney test; $: *p* < 0.05 significant.

**Figure 3 antibiotics-10-01205-f003:**
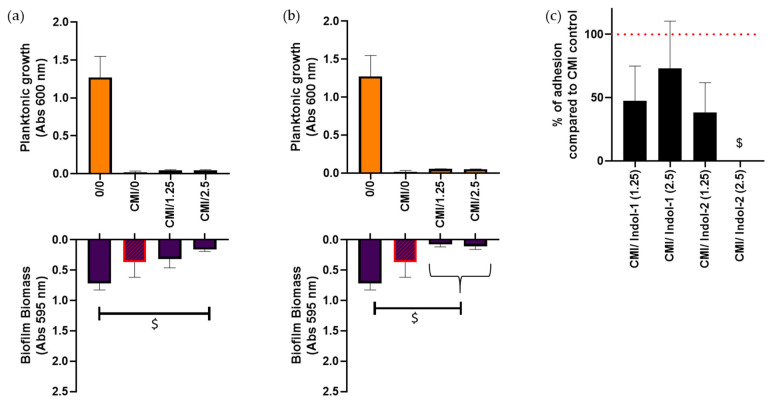
Ciprofloxacin MIC treatment combined with Indol-1 and Indol-2 molecules against *P. aeruginosa*. Ciprofloxacin concentration is annotated in first position and indoquinoline in second position in abscissa legend. (**a**,**b**) Planktonic growth and biofilm biomass after 24 h: (**a**) Indol-1 compound and MIC of ciprofloxacin (0.63 µg/mL); (**b**) Indol-2 compound and MIC of ciprofloxacin (0.63 µg/mL). MIC of ciprofloxacin appeared in red square. (**c**) Percentage of adherent cells after indoloquinolines + MIC of ciprofloxacin rationalized to the quantity of adherent cells after MIC of ciprofloxacin treatment alone. $: *p* < 0.05, significant vs. control.

**Figure 4 antibiotics-10-01205-f004:**
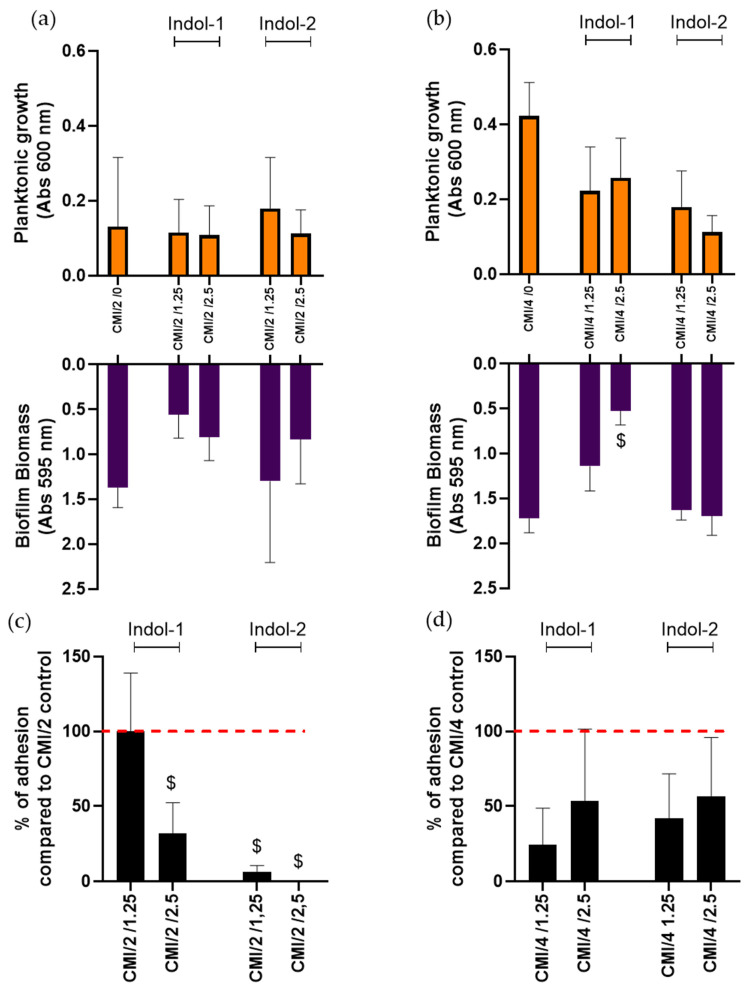
Sub-MIC Ciprofloxacin treatment combined with Indol molecules against *P. aeruginosa*. Ciprofloxacin concentration is annotated in first position and indoquinoline in second position in abscissa legend. (**a**,**b**) Planktonic growth and biofilm biomass after 24 h: (**a**) MIC/2 of ciprofloxacin (0.315 µg/mL); (**b**) MIC/4 of ciprofloxacin (0.1575 µg/mL). MIC ciprofloxacin appeared in red square. (**c**,**d**) Percentage of adherent cells after ciprofloxacin + Indol treatment rationalized to the quantity of adherent cells under ciprofloxacin treatment alone. (**c**) MIC/2 of ciprofloxacin. (**d**) MIC/4 of ciprofloxacin. Mann-Whitney test ($; *p* < 0.05, significant vs. control).

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
