# Peer review of "Synergy between Indoloquinolines and Ciprofloxacin: An Antibiofilm Strategy against Pseudomonas aeruginosa"

_antibiotics, 2021, doi:10.3390/antibiotics10101205_

Round 1
Reviewer 1 Report
The manuscript by Doudet et al describes the ability of synthetic coumpounds to enhance ciprofloxacin anti-biofilm activity towards Pseudomonas aeruginosa. The present study propose a combination of treatment to block P. aeruginosa biofilm formation. Interestingly from a therapeutic point of view, this strategy should make it possible to avoid the appearance of strain resistant to this combined treatment.
Finally, the topic and the data of this study are interesting and need just some experiments to improve the data.
Some complementary experiments and information’s should be done.
Major points Concerning the experiment realized:The toxicity of de two indoloquinoline compounds must be evaluated, using the concentration 1.25 and 2.5 µg/ml. For exemple using LDH survival test with A549 eukaryotic cells. It will be interesting to test a combination with both Indol-1 and Indol-2 associated together with ciprofloxacin, in order ti improve the anti-biofilm effect. An additional figure is mentioned (paragraph 2.2) but it is not present in the loaded document. Concerning methods:There is no information concerning the solubility of the molecules used? (Water, DMSO or other). In this case, how is the control prepared? This is to be described in the materials and methods
Minor point:
In materials and Methods section a S. aureus CIP 53.154 strain is mentioned. There is no data concerning this bacteria in the paper !! This is a mistake?
Author Response
Reviewer 1
The manuscript by Doudet et al describes the ability of synthetic coumpounds to enhance ciprofloxacin anti-biofilm activity towards Pseudomonas aeruginosa. The present study propose a combination of treatment to block P. aeruginosa biofilm formation. Interestingly from a therapeutic point of view, this strategy should make it possible to avoid the appearance of strain resistant to this combined treatment.
Finally, the topic and the data of this study are interesting and need just some experiments to improve the data.
Some complementary experiments and information’s should be done.
We thank the reviewer for comments for the improvement of our manuscript.
Major points
Concerning the experiment realized:The toxicity of de two indoloquinoline compounds must be evaluated, using the concentration 1.25 and 2.5 µg/ml. For exemple using LDH survival test with A549 eukaryotic cells.
We totally agree with the reviewer on the pertinence of cytotoxicity tests. Here, we wanted to underline the proof of concept in using non-antimicrobial molecules with antibiotic but these non-antimicrobial molecules will be modified to gain in improvement. In the context of the development of molecules possessing antiplasmodial activity, we have evaluated cytotoxicity of 10 µg/mL Indol-1 and Indol-2 compounds under physiological conditions associated with the use of circulating peripheral blood mononuclear cells (PBMCs and PMNsc). The study of mitochondriallactate deshydrogenase activity in cells supernatants and the measurement of DNA quantity allowed us to claimed that none of the compounds exhibited statistically relevant cytotoxic activity on circulating blood cells.
It will be interesting to test a combination with both Indol-1 and Indol-2 associated together with ciprofloxacin, in order ti improve the anti-biofilm effect.
Unfortunately this assay has to be post-ponded, it can be done right now as the quantities of compound left are too low, and molecules are being synthetized. Nevertheless, Indol-1 and Indol-2 are very similar molecules. We do hypothesis that they have the same mechanism of action and that using a higher concentration of molecules will lead to the same results. Both combination will be tested as soon as the compounds are ready.
An additional figure is mentioned (paragraph 2.2) but it is not present in the loaded document.
We apologize for this mistake and we add the file to revised documents.
Concerning methods:There is no information concerning the solubility of the molecules used? (Water, DMSO or other). In this case, how is the control prepared? This is to be described in the materials and methods
We added all these missing information in materials and methods. “Molecules were dissolved in DMSO at 1 mg/mL and then diluted in bacterial culture media (to 1/400 for 2.5 µg/mL concentration and 1/800 for 1.25 µg/mL concentration). » line 231-233
“Molecules were dissolved in DMSO at 1 mg/mL and then diluted in bacterial culture media (to 1/400 for 2.5 µg/mL concentration and 1/800 for 1.25 µg/mL concentration).”
Minor point:
In materials and Methods section a S. aureus CIP 53.154 strain is mentioned. There is no data concerning this bacteria in the paper !! This is a mistake?
We apologize for this mistake. We deleted this part.
Reviewer 2 Report
General comment:
An interesting original article discussing the synergical effect of the most common quinolone antibiotic and indoquinolines (indol-1 and indol-2). The results are promissing, but my recommendation for the future is to test other methods such as XTT-menadione or fluorescein diacetate (FDA).
Specific comments:
- Whenever used for the first time, even in the title, the name of the species is in full, followed by the initial in the brackets, like this: Pseudomonas aeruginosa (P. aeruginosa). Moreover, my opinion is that is inappropriate to use the initial in the title.
- For the biofilm biomass evaluation, crystal violet staining (CV) was used, but it would of been interesting to compare the results of this method with XTT-menadione assay, to evaluate both metabolic activity and biofilm.
Introduction:
- Lines 30-31: The phrase "Unfortunately, these super structures are tolerant to antibiotics and immune system representing a high concern for public health." should be rephrased. Maybe resistant instead of tolerant
Materials and methods:
- Lines 223-224: "S. aureus CIP 53.154 also named ATCC9144 or NCTC 6571 and P. aeruginosa CIP 82.118" My question is, if detailing S. aureus, with the type of ATCC strain, why not detail P. aerugionosa too, as ATCC 9027? In the end, this is the main subject of the paper.
Author Response
Reviewer 2
General comment:
An interesting original article discussing the synergical effect of the most common quinolone antibiotic and indoquinolines (indol-1 and indol-2). The results are promissing, but my recommendation for the future is to test other methods such as XTT-menadione or fluorescein diacetate (FDA).
We thank the reviewer for the interest to our paper and for advices.
Specific comments:
- Whenever used for the first time, even in the title, the name of the species is in full, followed by the initial in the brackets, like this: Pseudomonas aeruginosa (P. aeruginosa). Moreover, my opinion is that is inappropriate to use the initial in the title.
We modified according to reviewer’s comments (in the title and line 44).
- For the biofilm biomass evaluation, crystal violet staining (CV) was used, but it would of been interesting to compare the results of this method with XTT-menadione assay, to evaluate both metabolic activity and biofilm.
We totally agree with the reviewer and we will definitively plan this for the next generation of our molecules. We add a sentence in the discussion.
“Thus, investigations will be needed on the impact of this specific combination treatment on matrix biofilm but also on metabolic activity of bacteria within biofilm thank to XTT-menadione assay to decipher the mechanism of action involved.” Line 206-208
Introduction:
- Lines 30-31: The phrase "Unfortunately, these super structures are tolerant to antibiotics and immune system representing a high concern for public health." should be rephrased. Maybe resistant instead of tolerant
Resistance is a permanent state whereas tolerance is transitory. We believe that biofilms are not resistant to antibiotics or immune system as this state is transitory: released bacteria from biofilm are sensitive again. The structure of biofilm itself creates the tolerance, but bacteria themselves are not resistant.
Materials and methods:
- Lines 223-224: "S. aureus CIP 53.154 also named ATCC9144 or NCTC 6571 and P. aeruginosa CIP 82.118" My question is, if detailing S. aureus, with the type of ATCC strain, why not detail P. aerugionosa too, as ATCC 9027? In the end, this is the main subject of the paper.
We add this information on material and method section. “P. aeruginosa CIP 82.118 or ATCC 9027 is a referent strains cited in European Pharmacopeia.” Line 236
Reviewer 3 Report
This paper present a very interesting work about the use of two combined compounds, indoloquinoline derivatives and ciprofloxacin at MIC or sub-MIC concentrations to prevent P. aeruginosa biofilms.
While the subject of the present work is extremely interesting due to the increasing concerns regarding the antibiotic resistance, which represent a major health problem and highlight the urgent need of more studies on non-antimicrobial molecules used in combination with known antimicrobial molecules not only to avoid resistance but also for its worrying side effect,, there are some minor issues that need to be clarified before it may be accepted for publication.
- Some weakness in writing needs to be amended.
- The material is not presented in a proper way to allow easy comprehension.
- The introduction is not clear, need to have more data on compounds, resistance, and biofilm treated with them.
- Others anti-biofilm experiments with synergistic compound are recently reported in papers in which is described the importance to present antibiofilm properties independent from their antimicrobial activity. Please add in the introduction.
- It could be appropriate to detect also eradication capacity of the mixed compounds
- Despite the large amount of work, the authors should better discuss the collected results because in some points they generate confusion.
- Results and discussion require an important improvement especially in statistical analysis
Anyway, the manuscript is suitable for publication in Antibiotics after minor revisions.
Author Response
Reviewer 3
This paper present a very interesting work about the use of two combined compounds, indoloquinoline derivatives and ciprofloxacin at MIC or sub-MIC concentrations to prevent P. aeruginosa biofilms.
While the subject of the present work is extremely interesting due to the increasing concerns regarding the antibiotic resistance, which represent a major health problem and highlight the urgent need of more studies on non-antimicrobial molecules used in combination with known antimicrobial molecules not only to avoid resistance but also for its worrying side effect,, there are some minor issues that need to be clarified before it may be accepted for publication.
We thank the reviewer for these positive comments.
- Some weakness in writing needs to be amended.
We made correction all along our article.
- The material is not presented in a proper way to allow easy comprehension.
We modified the section of material and methods for an easier comprehension.
- The introduction is not clear, need to have more data on compounds, resistance, and biofilm treated with them.
We add information in the introduction “To our knowledge, antibiofilm activity of these compounds or the emergence of resistance against them were not tested.” Line 55-56
- Others anti-biofilm experiments with synergistic compound are recently reported in papers in which is described the importance to present antibiofilm properties independent from their antimicrobial activity. Please add in the introduction
We added this information in introduction and included these references.
“This approach has already been tested with promising results underlining the importance to independently present antimicrobial and antibiofilm activities of molecules (15-18). » Line 58-60
- It could be appropriate to detect also eradication capacity of the mixed compounds
We agree with the reviewer’s comment. Our first objective was to test these compounds for a preventing action, to avoid biofilm formed under ciprofloxacin CMI. But it will be necessary to test these compounds for their biofilm eradication action. We added a sentence in the discussion.
“Moreover, indoloquinolone-ciprofloxacin combined drugs will have to be tested for their biofilm eradication capacity.” Line 207-208
- Despite the large amount of work, the authors should better discuss the collected results because in some points they generate confusion.
We modified different point in the discussion to avoid confusion.
- Results and discussion require an important improvement especially in statistical analysis
We modified different points in the text to improve our statistical analysis
Round 2
Reviewer 1 Report
The authors responded to my questions and requests satisfactorily
Author Response
Concerning the experiment realized: The toxicity of de two indoloquinoline compounds must be evaluated, using the concentration 1.25 and 2.5 μg/ml. For exemple using LDH survival test with A549 eukaryotic cells.
- Please add the requested experiment to your study or provide information on cytotoxic activities of these compounds in the manuscript.
We added a sentence at the end of first paragraph of results “In the context of another study, none of the compounds (10 µg/mL Indol-1 and Indol-2) exhibited statistically relevant cytotoxic activity on circulating peripheral blood mononuclear cells.”
It will be interesting to test a combination with both Indol-1 and Indol-2 associated together with ciprofloxacin, in order ti improve the anti-biofilm effect.
- Please add the experiment to your study or elaborate on this briefly within your manuscript
We added a sentence at the end of discussion “Another approach will be to combine Indol-1 and Indol-2 associated together with ciprofloxacin, in order to improve anti-biofilm effect.”
Concerning methods:There is no information concerning the solubility of the molecules used? (Water, DMSO or other). In this case, how is the control prepared? This is to be described in the materials and methods
- Please add information on the preparation of controls (e.g. non-treated, 0 ug/ml controls)
We apologize for this oversight. We add this information in method section. “In each experiment, a solution with the same amount of DMSO but without any molecule was prepared and tested as a control.”
